# Changes in Chromatin Organization Eradicate Cellular Stress Resilience to UVA/B Light and Induce Premature Aging

**DOI:** 10.3390/cells10071755

**Published:** 2021-07-11

**Authors:** Bela Vasileva, Dessislava Staneva, Natalia Krasteva, George Miloshev, Milena Georgieva

**Affiliations:** 1Laboratory of Yeast Molecular Genetics, Institute of Molecular Biology, Bulgarian Academy of Sciences, 1113 Sofia, Bulgaria; belavas@outlook.com (B.V.); dessysta@gmail.com (D.S.); karamolbiol@gmail.com (G.M.); 2Institute of Biophysics and Biomedical Engineering, Bulgarian Academy of Sciences, 1113 Sofia, Bulgaria; natalia.krasteva@yahoo.com

**Keywords:** *Saccharomyces cerevisiae*, aging, chronological aging, actin-related protein 4, linker histone, chromatin, UVA-B irradiation, stress resilience

## Abstract

Complex interactions among DNA and nuclear proteins maintain genome organization and stability. The nuclear proteins, particularly the histones, organize, compact, and preserve the stability of DNA, but also allow its dynamic reorganization whenever the nuclear processes require access to it. Five histone classes exist and they are evolutionarily conserved among eukaryotes. The linker histones are the fifth class and over time, their role in chromatin has been neglected. Linker histones interact with DNA and the other histones and thus sustain genome stability and nuclear organization. *Saccharomyces cerevisiae* is a brilliant model for studying linker histones as the gene for it is a single-copy and is non-essential. We, therefore, created a linker histone-free yeast strain using a knockout of the relevant gene and traced the way cells age chronologically. Here we present our results demonstrating that the altered chromatin dynamics during the chronological lifespan of the yeast cells with a mutation in *ARP4* (the actin-related protein 4) and without the gene *HHO1* for the linker histone leads to strong alterations in the gene expression profiles of a subset of genes involved in DNA repair and autophagy. The obtained results further prove that the yeast mutants have reduced survival upon UVA/B irradiation possibly due to the accelerated decompaction of chromatin and impaired proliferation. Our hypothesis posits that the higher-order chromatin structure and the interactions among chromatin proteins are crucial for the maintenance of chromatin organization during chronological aging under optimal and UVA-B stress conditions.

## 1. Introduction

As cells age, their stress resistance reduces. This accounts for different cellular damages, finally leading to death [1,2]. Cellular aging includes replicative and chronological aging, both accounting for the accumulation of damages at the molecular, cellular, and organismal level [3]. Replicative aging is expressed in the number of mitotic divisions of a cell, while chronological aging is the time during which a cell has stopped dividing, but still preserves its life [4]. The yeast *Saccharomyces cerevisiae* is considered as a benchmark for studying aging [5,6]. On one hand, yeast cells are unicellular and easy to work with, while on the other, they share many similarities to human cells. All main signaling pathways and key biological processes implicated during aging as the cell cycle impairment, the increased cell death, the alterations in the vesicular transport, the changed metabolism, the detected protein misfolding, and degradation share evolutionarily conserved patterns [4,7,8]. Moreover, *S. cerevisiae* is the first eukaryotic organism to have its genome sequenced, which brings a significant contribution to aging research [9]. Another main feature of baker’s yeast cells is that they allow clear boundary between the replicative and chronological aging, something impossible in higher eukaryotes. This exempts the opportunity to look separately at the main features of the two ways eukaryotic cells age [10]. Furthermore, certain protocols are allowing the follow-up of the complete chronological lifespan (CLS) for less than a month, something that is impossible in higher eukaryotes. As yeast cells grow, around the 24th cultivation hour they enter a post-diauxic phase, which is followed by exiting of the cell cycle by the majority of cells [11]. During this period, they experience a growth reduction and their metabolism switches to a mitochondrial respiratory mode. Reaching the end of the post-diauxic phase, the cells enter the stationary phase, between day 2 and 7, depending on the medium, where they cease to divide and only sustain their viability [4]. This gives possibilities to follow the overall metabolic switch during the lifespan and study certain factors that control this. Recently there has been a huge amount of research on yeast CLS aimed at using these unicellular eukaryotes as a model for testing the antiaging activities of different compounds [12,13] and for studying the fine molecular mechanisms that underlie the process, including research on cellular metabolism and telomere biology in regard to aging [14,15]. Moreover, personal perspectives also appeared highlighting the importance of the yeast CLS research as a compelling stipulation for providing really deep insights on the process in higher eukaryotes [16]. Interestingly, some authors have provided insights on the role of mitochondria and its metabolisms in the process of yeast CLS [17], thus presenting evidence that the functional state of mitochondria is vital to cellular and organismal aging in eukaryotes across phyla. Others highlighted the importance of the communications between mitochondria, the nucleus, the vacuoles, the peroxisomes, the endoplasmic reticulum, the plasma membrane, the lipid droplets, and the cytosol during yeast CLS [16,18].

The communication and interaction among all cellular compartments and systems during aging is of great importance. Therefore, studies that link the way how cellular compartments communicate during aging are indispensable. Chromatin plays crucial role in aging [19,20]. Chromatin remodeling by either addition or removal of histones, or by the effect of different post-translational modifications such as methylation, acetylation, ubiquitination, and phosphorylation proves determinative for the way cells will age with time [21]. In aging and age-associated diseases like cancer, there is a switch in the epigenetic pattern, which itself is important for normal development [22]. Progeria diseases, for example, Hutchinson-Gilford progeria syndrome (HGPS), characterized by a premature aging phenotype, experience a disruption of the nuclear organization and a disordered chromatin structure. Such changes are exhibited in normally aging cells, while in HGPS they occur at a much more accelerated rate. This gives another evidence for a link between aging and disrupted chromatin structure [23,24]. As essential counterparts of the higher-order chromatin organization, the linker histones are important players in aging and stress adaptation [25,26,27].

One of the main reasons for aging is the production of free oxygen species especially during prolonged exposure to ultraviolet (UV) rays. The process is called photoaging and has been proven to be damaging to the skin, as well as leading to epidermal stem cell exhaustion [28]. There are three types of UV rays—UVA (320–400 nm), UVB (290–320 nm), and UVC (190–280 nm). The most damaging, UVC, is blocked by the ozone layer and therefore cannot reach the Earth’s surface [29]. Even though invisible to the naked eye, as they are absorbed by the skin, this light causes severe DNA damage, inhibiting DNA replication and transcription and promoting genome instability. They work by producing photo lesions, which can result in incorrectly incorporated nucleotides in the newly synthesized DNA [30]. Most of the UVB radiation is absorbed by the upper epidermal layer. It causes DNA damage by forming dimeric photoproducts like cyclobutane pyrimidine dimers and pyrimidine (6-4) pyrimidone photoproducts [31]. UVB is the main cause of sunburn and most types of skin cancer. In contrast, UVA reaches deeper—to the basal layer of the epidermis and is the main initiator of oxidative damage, as well as more permanent aging features like wrinkles, and some types of skin cancer [32].

*Saccharomyces cerevisiae* cells, being the type of organism that allows differentiation between replicative and chronologically aging cells, gives a perfect opportunity for studying the relationship between these two types of aging under stress conditions. Hence, we aimed at examining the different molecular mechanisms of cellular response to UVA/B light stress during the process of chronological aging in yeast cells with different chromatin backgrounds. For this purpose, we have used three chromatin mutants—cells that lack the gene for the linker histone, a strain that has a point mutation in *ARP4* (the actin-related protein 4) gene and a double mutant with the two mutations induced at the same time [33] and compared their ability to survive UVA/B stress. Moreover, we have studied the fine molecular mechanisms that underlie the way cells chronologically age under normal and UVA/B stress conditions regarding their cell proliferation, gene expression and chromatin organization. Our results demonstrate that the altered chromatin dynamics during the chronological lifespan of these yeast cells leads to strong alterations in the gene expression profiles of a subset of genes involved in DNA repair and autophagy, reduced survival upon UVA/B irradiation, and impaired proliferation.

## 2. Materials and Methods

### 2.1. Yeast Strains

Wild type (WT)—*MAT**a** his4-912δ-ADE2 his4-912δ lys2-128δ can1 trp1 ura3 ACT3 [34];*

*hho1Δ *(in the text appears as *hho1delta*)—*MAT**a** his4-912δ-ADE2 his4-912δ lys2-128δ can1 trp1 ura3 ACT3 ypl127C::K.L.URA3 [34,35];*

*arp4ts26* (designated in the text as *arp4*)—*MAT**a** his4-912δ-ADE2 lys2-128δ can1 leu2 trp1 ura3 act3-ts26 [34];*

*arp4ts26Δhho1* (denoted as *arp4 hho1delta*)—*MAT**a** his4-912δ-ADE2 lys2-128δ can1 leu2 trp1 ura3 act3-ts26 ypl127C::K.L.URA3 [35];*

The *arp4* mutant is a conditional mutant harboring a temperature-sensitive allele of *ACT3*. A permissive temperature for growth for these cells is 23 °C, while 37 °C is lethal. At 30 °C, the temperature at which the strains were cultivated in our experiments, the mutant propagated and the mutation was expressed [34].

### 2.2. Chronological Life Span (CLS)

The CLS of *S. cerevisiae* was examined according to [4,27,36]. Briefly, the yeast strains were grown in SD media, containing 1.7% yeast nitrogen base, 20 μg/mL of all required amino acids according to the reported genotype of the strains and 2% dextrose. Cell cultivation continued for 9 days at optimal conditions (30 °C) in a water bath shaker. At certain time points, namely the 4th, 24th, and 72nd h, and at the 9th day of cultivation, cell aliquots were spectrophotometrically measured. Results of the OD_600_ (optical density at λ 600 nm) were used for building growth curves.

### 2.3. UVA/B Irradiation of Cells and Chronological Lifespan (CLS)

Yeast cells were cultivated in complete minimal medium (SD) and aliquots of them were taken at the 4th, 24th, and 72nd h and at the 9th day of cultivation. One hundred cells were spread on a solid YPD (1% yeast extract, 2% peptone, and 2% dextrose) media. Cells were then irradiated with UVA/B light for 3 or 30 min. The used source of UVA/B was a 15W Cleo lamp, with the help of which an energy dose of 2.56 mW/cm^2^ was obtained. The irradiation was followed by a 2-day recovery period at 30 °C on a rich YPD plate. After this time, the number of colonies was counted and the percentage of viable cells was estimated by taking the non-irradiated controls as 100%. Three repetitions of the experiment were performed and results are presented as MEAN ± STDV.

### 2.4. Chromatin Yeast Comet Assay (ChYCA) for Higher-Order Chromatin Loop Organization Studies

Control and UVA/B irradiated yeast cells at the four monitored time points of the CLS were studied by Chromatin Yeast Comet Assay (ChYCA) according to the protocol developed by Georgieva et al. [37,38]. Briefly, cells were washed in 1× PBS (2.68 mM KCl, 1.47 mM KH_2_PO_4_, 1.37 mM NaCl, 8 mM Na_2_HPO_4_; pH 7) and resuspended in S-buffer (1M Sorbitol; 25 mM NaH_2_PO_4_, pH 6.5) to a concentration of 1 × 10^5^ cells/mL. After solidifying the gels for 5 min at 10 °C the coverslips were removed followed by in situ treatment with 200 U/mL of micrococcal nuclease (MBI Fermentas) diluted in micrococcal buffer (1 M Sorbitol, 5 mM CaCl_2_, 10 mM Tris-HCl, pH 8.0, 5 mM NaCl). The gels were then covered with coverslips and incubated at 37 °C for 3 min. The enzyme reaction was stopped by immersing the slides into lysis solution (146 mM NaCl; 30 mM EDTA; 10 mM Tris-HCl and 0.1% *N*-lauroylsarcosine, pH 7.5) for 20 min in a cold room at 10 °C. Afterwards slides were washed for 15 min in 0.5× TBE buffer (44.5 mM Tris, 44.5 mM Boric-acid, 2.5 mM EDTA, pH 8). Electrophoresis followed for 10 min at 0.45 V/cm in the same TBE buffer under neutral conditions and slides were dehydrated and visualized under a fluorescent microscope Leitz (Orthoplan, VARIO ORTHOMAT 2) using 450–490 nm bandpass filter after staining with SYBR green. Pictures were taken with a build-in microscope photo camera. CometScore performed results quantitation and data were presented as “comet length” in arbitrary units. Three repetitions of the experiment were performed and results are presented as MEAN ± STDV.

### 2.5. Fluorescence-Activated Cell Sorting (FACS) Analysis

FACS analysis was performed according to [37]. Several changes were made: control and irradiated cells were fixed with 100% ethanol; sonication lasted for 20 s at 50% power; cells were incubated with RNase A (0.25 mg/mL), after which they were centrifuged for 4 min/5000 g. A pre-staining with propidium iodide was followed by BD FACSCanto apparatus monitoring. Quantification of the results was made by FlowJoV10 software. Three repetitions of the experiment were performed and results are presented as % of the whole population of cells.

### 2.6. Reverse Transcription Quantitative PCR (RT qPCR)

Total RNA of 5 µg was obtained for the four yeast strains, both before and after irradiation for either 3 or 30 min, at the 4th, 24th, and 72nd hour and at the 9th day of cultivation. The RNA was then reverse transcribed to cDNA. Four genes of interest—*HHO1, RAD9, ATG18,* and *CDC28*, and one reference gene *ACT1* were analyzed. The following primers were used:*HHO1*forward—5′-TTACCGCCAAGAAGGCCTCTTC-3′reverse—5′-ATACGGCTGGAGCCCTTACCG-3′*RAD9*forward—5′-GCATGTTTGAGCGCAGGTAG-3′reverse—5′-TCTGGGTACTAAAGAATCTAAGGCA-3′*ATG18*forward—5′-TTCCCGTTGAAACCAATTCCCA-3′reverse—5′-GCCAGTTTCGAAGAGTTCCGGAT-3′*CDC28*forward—5′-AGGAAACCAATCTTCAGTGGCGA-3′reverse—5′-CTGGCCATATAGCTTCATTCGGC-3′*ACT1*forward—5′-CGGTAACATCGTTATGTCCGGTG-3′reverse—5′-ATGGAAGATGGAGCCAAAGCG-3′.

The ΔΔC_T_ method was used for the calculation of the results. Three repetitions of the experiment were done.

### 2.7. Statistical Analyses

All experiments were done in three repetitions and results were statistically elaborated with Excel.

## 3. Results and Discussion

### 3.1. The Mutant Yeast Cells, Lacking Functional Arp4p and the Gene for the Linker Histone, Exhibit Slower Growth

Four different *S. cerevisiae* strains—a wild type and three chromatin mutants were cultivated under optimal and UVA/B stress conditions. The *hho1delta* mutant has a deleted gene for the linker histone, *HHO1* [34], the *arp4* mutant has a point mutation in the gene for Arp4p [34], while the double mutant *arp4 hho1delta*, combines the two mutations [35]. The strains were selected from our previous work [33] to allow monitoring of the chronological lifespan in different chromatin backgrounds under UVA-B stress and our previous results have shown that the three chromatin mutants have defects in their chromatin loop organization [26,35] and impaired, accelerated chronological aging. Moreover, to further discuss the interaction between the yeast linker histone Hho1p and Arp4p we have to point to the fact that *ACT3/ARP4* is an essential gene, coding for the actin-related protein Act3p/Arp4 of *Saccharomyces cerevisiae* located within the nucleus. Act3p/Arp4 is a component of the NuA4, INO80 and SWR1 chromatin modulating complexes, and recruits these complexes onto chromatin for their proper chromatin functions. Studies, including ours, have shown that mutated Act3p/Arp4 led to impairment of the functions of these remodeling complexes, and affected chromatin organization and function [33,34,38].

In the current study, we have monitored the way these chromatin mutants age under UVA/B irradiation and checked how chromatin organization influenced the process of CLS under optimal and UVA/B stress conditions. The main idea was to follow how the primary chromatin compaction impacted the way cells age and potentiated their ability to withstand UVA/B stress. For this purpose, as a first task the cell culture growth of the four yeast strains was followed for nine consecutive days under optimal cultivation conditions without stress application. The purpose was to have information on how cells grow under normal conditions without additional stress. These data were prerequisite for all following yeast CLS experiments as they gave the foundations of the whole work afterward. Our previous studies with the CLS of the four strains published in [26] have investigated the survival and aging of the same yeast cell cultures for a period of 19 days. These results have demonstrated that at day 9 we already had prematurely aged cells among the double mutant, while the *hho1delta* mutants exhibited better survival under optimal CLS conditions than all other strains. Therefore, we deliberately chose to monitor the CLS of the four strains in the current study until day 9. This allowed us to study the CLS under UVA/B stress conditions without losing most of the cells in the double mutant as already prematurely dead as shown in [26].

Overnight cultures were transferred into a complete minimal medium (CMM) at a density of 1 × 10^7^ cells/mL. Cells were then left to age in a shaker incubator at 30 °C. Spectrophotometric analyses (OD_600_) were performed by measuring the density of the cultures at the studied four different time points: 4th, 24th, and 72nd h and at the 9th day of cultivation. The obtained results are represented as growth curves in Figure 1. The arrow line below the graph with the studied time points designates the CLS follow-up period. Broadly accepted in the literature is the fact that around the 24th hour, the yeast cells enter the stationary phase and begin to age chronologically [10]. The chromatin mutants exhibited a noticeable change in their growth, compared to the control (green line). The double mutant *arp4 hho1delta* showed lower OD_600_ compared to all other strains with the explicit drop in the OD_600_ on the last day. The double mutant *arp4 hho1delta* showed an OD_600_ reduction of 62% (versus WT, 9th day), followed by a reduction in the cell growth of the *hho1delta* mutant with 36%, and the *arp4* mutant with 16%. As noted by previous results obtained in our laboratory [26], the chromatin mutants experienced a lower cell growth compared to the wild type, and especially those that lack the linker histone gene. As also shown here, the most pronounced reduction in the yeast cell culture growth was observed for the *arp4 hho1delta* mutant, where the combination of the two mutations showed a synergistic cell growth inhibitory effect.

The four strains were studied for a period of nine days in 2% dextrose SD media. At four time points (4th, 24th, and 72nd h, and on day 9) the optical density (OD_600_) was measured by a spectrophotometer at λ 600 nm. Three repetitions of the experiment were done and the results are presented as MEAN ± STDV. The green arrow demonstrates the way the studied yeast cells propagated through the different time points monitored under the study with the switch between the logarithmically growing and stationary phase cells designated with red arrows.

### 3.2. The Lack of the Linker Histone and the Abolished Interaction with Arp4p Alter Chromatin Loop Dynamics during the CLS

There is an extensive interest in the way chromatin governs the chronological aging of eukaryotic cells. Data exist that prove its fundamental role [39]. It has been confirmed that during organismal aging, cells experience heterochromatin loss, DNA demethylation and global histone loss [40]. Moreover, alterations in nuclear organization and function are pronounced upon physiological and mitochondrial stress and during premature and physiological aging [20,27,41,42]. Regarding the higher-order chromatin organization and its involvement in aging several groups have reported the role of lamins in aging [23,43], thus insisting on detailed studies on the role of higher-order chromatin dynamics in aging [27,44]. Our previous research has studied the way higher-order chromatin organization changes during CLS in yeast mutants that lack the gene for the linker histone and results are published elsewhere [27].

To monitor the way four studied yeast strains remodel their global chromatin compaction during chronological aging we performed Chromatin Yeast Comet Assay (ChYCA) in which cells at the four time points were subjected to micrococcal nuclease (MNase) digestion and then electrophoresed under mild conditions [37,38]. The last allowed digested chromatin loops to protrude toward the anode and to form comet tails. Data quantitation was carried out by calculating the length of the observed comets with CometScore and results are presented in Figure 2. The control WT cells showed the smaller increase in their comet tails during CLS, while the mutants lacking the gene for the linker histone (*hho1delta* and *arp4 hho1delta*) had the highest rate in chromatin decompaction during the CLS with the most explicitly increased comet tails’ lengths on the 24th h and day 9. The detected increase in the comet tail lengths was with 40% and 60% for the mutant *hho1delta,* compared to its comet tails on the first time point. For the double mutant an increase of 26% compared to the WT was detected earlier at the first time point and continued until the last when no comets were detected. This disappearance of comets could be explained by the fact that at this time point there were no more viable yeast cells, a result of complete digestion of chromatin by MNase and lack of DNA to form comets. The other chromatin mutant *arp4* also demonstrated decompaction of chromatin at the first time point, which at the next two time points (24th and 72nd h) was insignificant. On day 9 these cells had longer comet tails like the mutant lacking the gene for the linker histone, suggesting disorganized chromatin with higher accessibility for MNase. These results represent the accelerated dynamics in chromatin decompaction during CLS for the three chromatin mutants and highlight the role of the linker histone for maintaining chromatin organization during aging. To the best of our knowledge, this is the first chromatin assay for monitoring the way chromatin organization disorganizes during aging in yeast mutants that lack the gene for the linker histone and with a mutation in *ARP4*. Our previous studies have shown that the cells lacking only the gene for *HHO1* had a higher chromatin decompaction resulting in premature aging [27]. The results from the current study prove that the double mutants experience accelerated chromatin decompaction during CLS, that at the last time point resulted in a complete lack of chromatin loop organization, i.e., of comets.

### 3.3. Yeast Mutants with Mutated arp4 and Lacking the Gene for the Linker Histone (arp4 hho1delta) Show Signs of Premature Aging under Low- and High-Dose UVA/B Stress

The resilience to UVA/B irradiation of the studied cells was further probed. UVA/B light has a tremendous effect on how our skin ages, by causing photo-aging [45]. It is also a major contributor to the development of different skin cancers, as it promotes genome instability [46]. Knowing how it affects the growth of actively dividing and non-dividing cells in the process of chronological aging for the wild type and the studied chromatin mutant strains, will bring us significant insights into the molecular mechanisms of the cellular response to stress and the importance of chromatin structure for this process.

WT, *hho1delta*, *arp4*, and *arp4 hho1delta* were cultivated in CMM. At the four time points: the 4th, 24th, 72nd h and 9th day of cultivation, aliquots were taken and an equal number of cells, 10^2^, were plated on solid YPD petri dishes. For all four strains, at the four different time points, we had a control and a UVA/B treated sample. The doses were chosen as representatives of very moderate sun exposure and higher sunlight exposure, similar to the UVA/B doses used in the tanning bed industry. For 3 min irradiation, the calculated dose was 460.8 mJ, whereas for 30 min—4608 mJ. The ability of all four strains to form colony-forming units (CFU) was measured for both the low- and high-dose irradiated cells (Figure 3). The results, after irradiation with the lower UVA/B dose, are shown in Figure 3A. The observation of the trends of the three mutants’ cell survival after this UVA/B irradiation showed a slight stimulating effect of their CLS after the 24th hour of cultivation compared to the WT. However, at the 72nd hour *hho1delta* and *arp4* mutants had an induction in their viability with the mutant without the linker histone (*hho1delta*) explicitly showing much higher cell viability after UV stress. The linker histone H1 is considered as an inhibitor of homologous recombination [47].

Therefore, the lack of it could result in an abnormal repair of DNA strand breaks and thus to increased UVA/B survival at this dose. For the *arp4* mutants, Arp4p is a main subunit of the INO80 chromatin-remodeling complex. INO80 mutant *Saccharomyces cerevisiae* cells possess an increased removal of cyclobutane pyrimidine dimers induced by UV light [48]. Interestingly, the combination of these two mutations in the double mutant did not result in an increase of viability. In contrast, a sharp decrease in *arp4 hho1delta* double mutant cells’ viability was detected at all-time points with no growth on the 9th day of cultivation. This trend suggests premature aging of the double mutant cells under 460.8 mJ UVA/B irradiation and is in unison with previous results showing premature aging in these cells under normal conditions [26].

The results after irradiation with higher doses of UVA/B light (Figure 3B) demonstrated the inability of the three chromatin mutants to withstand high UVA/B irradiation, resulting in generally reduced viability, especially visible for the double mutant, which again had non-dividing cells on the 9th day of cultivation

### 3.4. The hho1delta and arp4 hho1delta Mutant Cells Demonstrate an Impaired Cell Cycle Progression under Optimal CLS

The CLS studies in yeast are of major importance as they allow detailed analyses of cellular metabolism, cell cycle progression, and the transition between the logarithmically growing and stationary phase cells, as well as between quiescent and non-quiescent stationary phase cells [49]. Apparently, when yeast cells enter the stationary phase after the 24th h of CLS cultivation in 2% dextrose SD media certain proportion of the cells are in a quiescent state while some are in a non-quiescent state [50]. Though certain techniques exist for their separation and analysis, little is known for the underlying mechanisms for this diversity in the properties of stationary phase cells from the same yeast strain. Moreover, recent data have linked the yeast CLS with perturbation in the cell cycle which reflected the proportion of quiescent and non-quiescent cells under calorie restriction [50]. In order to shed light on the cell cycle progression of the four studied strains we performed FACS cell cycle analysis at the four chosen time points: 4th, 24th, 72nd h and 9th day of cultivation. This allowed us to compare the difference in the proliferation of actively replicating cells (4th and 24th h time points) to those that are in a chronological lifespan (72nd h and 9th day), in other words, to compare young to old cells.

The results of this experiment are shown in Figure 4. A clear trend was observed on the 4th hour of CLS where the majority of *hho1delta* and *arp4 hho1delta* mutants were in the G0/G1 phase of the cell cycle (Figure 4A). This was contrasting to the other two strains—WT and *arp4*, in which the bulk portion of cells was in the G2/M phase, something that has been reported as a typical feature for actively dividing young cells [51]. These differences in the way cells propagated through the cell cycle phases in the mutants that lack the gene for the linker histone stayed constant regardless of the measured time point (Figure 4A–D). Built-in representative FACS data cell cycle histograms are presented for each time point and they allow the visual comparison of the different behavior of the four studies’ yeast strains when regarding the propagation in the phases of the cell cycle. It was obvious that from the initial time point 4th h onward *hho1delta* mutant cells did not show any signs of prematurely aged cells, probably due to escaping the post-diauxic shift or by staying in a non-quiescent state. Moreover, when we refered to cell survival CLS data it was obvious that these mutants were pertaining their viability which reflected in their propagation through the cell cycle. In any case, in the later time points, all four strains had their cells predominantly in the G0/G1 phase, something that was accepted as typical for aged cells and an aging cell population [7]. The observed slight block of the cells in the G2/M phase of the cell cycle for the *hho1delta* and the *arp4 hho1delta* mutants demonstrated a premature aging phenotype, which was easily monitored on the last time point, namely the 9th day, for the mutant that lacked only the gene for the linker histone. Referring to the CLS, i.e., the colony-forming units of the four strains on day 9, we decided that the proliferation of cells seen at this later stage of the CLS for the double mutant was compromised as almost all cells of this mutant (2/3 versus WT and 50% versus 72nd h) were non-dividing at this point (Figure 1). Therefore, the 9th d-time point data cannot be evaluated clearly and needs further studies.

### 3.5. UVA/B Stress Impairs the Cell Cycle in All Studied Chromatin Mutants (arp4, hho1delta, and the Double Mutant Cells) at Earlier Time Points of CLS

Next, we performed FACS cell cycle examinations of the four yeast strains after irradiation with both low and high doses of UVA/B light (Figure 5). Cells were cultivated in CMM and at the four time points aliquots were taken for both control and irradiated samples. After UVA/B light exposure, both the irradiated and non-irradiated cell aliquots were returned to the shaking incubator at 30 °C for 90 min to recover, after which they were fixed with 95% ethanol and stored at −20 °C for FACS analysis. FACS analyses followed on the next day where the propagation of cells through the cell cycle phases was represented as cells in each phase of the cell cycle calculated as a percentage of the non-irradiated cells at the same time point for each strain. This representation of the results demonstrated the way the cells transit through the cell cycle phases in response to the applied UVA/B stress and allowed comparison with their non-irradiated sample. For the lower doses of UVA/B light (Figure 5A,C), regarding the young cells, all four strains demonstrated a decrease in the percentage of cells in the G2/M phase of the cell cycle and an increased proportion of cells in G0/G1 and S phase after exposure to 460.8 mJ UVA/B irradiation, mainly visible in WT, *arp4* and the double mutant (see 24th h-time point in Figure 5C).

An easily seen difference in the proliferation potential was observed for the *hho1delta* and the double *arp4 hho1delta* mutants at the 4th h-time point, where a slight increase in the cells in G0/G1 was detected after irradiation, compared to the controls. The trend was similar to the way these cells proliferated under optimal conditions (see Figure 4). During the other time points the cells, lacking the gene for the linker histone had a drop in G0/G1 cells and an increase in S and G2/M. At the last time point, the double mutant demonstrated an increase of cells in G0/G1 and a well displayed misregulated cell cycle as seen from the blurred built-in histograms, indicating an inability of cells to transit through the cell cycle phases. The other chromatin mutants kept the same tendency until the last time point. The exposure to higher doses of UVA/B light (4608 mJ) is shown in Figure 5B,D. The time point indicating replicating cells (4th h) showed that this UVA/B irradiation dose impaired the proliferation rate of all strains with a reduced portion of cells in all phases. The last was in unison with the viability results where this UVA/B dose impaired cellular viability significantly for all strains earlier than the lower one (see Figure 3). At later time points of CLS, the same tendency was followed. The same is easily seen by the representative built-in histograms presented for the two energy doses used for irradiation (Figure 5C,D).

### 3.6. The Expression Profiles of HHO1 and Some Stress-Responsive Genes Show a Correlation with the Way Cells Age

Since aging is associated with major changes in chromatin, and vice versa, as well as with histone and heterochromatin loss [20,44], it was of interest for our work to see the dynamics of *HHO1* (the gene for the *S. cerevisiae* linker histone [33]) expression in the timeline of yeast CLS under UVA/B stress. Moreover, other authors’ work implied that the linker histone H1 is lost in senescent cells correlating with the formation of senescence-associated heterochromatic foci (SAHFs) [52]. Therefore, the chromatin mutant yeast strains studied in this work were investigated for changes in the expression profiles of *HHO1* (in WT and *arp4* mutant) as well as the expression profiles of certain stress-responsive genes. For the purpose, cells were left to chronologically age and at the tested time points were irradiated with low and high doses of UVA/B (460.8 and 4608 mJ). Aliquots of non-irradiated and UVA/B irradiated cells were taken and were analyzed for changes in the expression profiles of the studied genes via RT-qPCR (reverse transcription-quantitative PCR). The results from the RT qPCR were calculated by the means of the ΔΔCT method, with the use of a reference gene *ACT1*. Rotor-Gene 6000 Series Software 1.7 was used for data calculations. The expression of four different genes was examined—*HHO1*, *RAD9*, *ATG18*, and *CDC28*, both before and after irradiation with UVA/B light.

The first examined gene was *HHO1* (Figure 6), as it encodes the yeast linker histone orthologue [53]. Its expression was probed in WT and *arp4* mutant. The interest in this gene came from its key role in the organization of the higher-order chromatin structure [27,37]. Its knockout results in impaired higher-order chromatin structure. Moreover, it is of great importance for the proper formation of chromatin loops, and its knockout leads to loss of the chromatin loop organization [37]. Our results showed that in the WT there was a decreased expression of the *HHO1* gene, after irradiation with lower doses of UVA/B, which gradually increased with the cells entering the stationary phase (Figure 6A). No significant changes were noted in the *arp4* mutant after irradiation with this UVA/B dose. This was changed when the dose was increased to 4608 mJ (Figure 6B). In general, the WT, except on the 72nd h, showed an increased expression of *HHO1* mRNA after irradiation, while the chromatin mutant showed a decreased expression that continued throughout the overall period of monitored CLS.

The second examined gene was *RAD9*, which is essential for exiting G2/M cell cycle arrest as it acts as a regulator of DNA damage [54]. Its gene expression during the CLS of the studied yeast strains is shown in Figure 7. For the treatment with the lower dose of UVA/B light (Figure 7A), no significant changes were observed in the chromatin mutants, compared to the WT after irradiation with an exception for the last time point where all strains had an increase in their gene expression level (Figure 7A). Interestingly, the double mutant *arp4 hho1delta* cells showed the lowest RAD9 expression, suggesting a link with the impaired chromatin organization. Non-irradiated cells had a decrease in RAD9 expression levels, explicitly observed on the 72nd h of CLS.

Therefore, the increase in the RAD9 expression levels after UVA/B irradiation with 460.8 mJ resulted in a boost of its expression and points to a cell cycle stress in all chromatin mutants. The results were different when the UV light dose was increased, resulting in a completely changed picture for the expression of *RAD9* in the chromatin mutants, compared to the WT (Figure 7B). The cells of the three chromatin mutants, due to their abrogated chromatin structure, could not cope with the exerted stress and a severe downregulation of *RAD9* was observed, particularly in the later stages of their development.

The third examined gene was *ATG18* (Figure 8), which is a part of the autophagy-related genes, required for vesicle formation [39]. Autophagy itself is a process of cellular “self-eating” as its purpose is to recycle damaged organelles, unfolded proteins, etc., without forcing the cell to go under apoptosis. It is characterized by the formation of the so-called autophagosomes and later autophagolysosomes, where the damaged cellular components are degraded [40]. As a result of the lower doses of UVA/B light (Figure 8A), there was a general decrease in the mRNA levels of *ATG18*, particularly for the stationary phase (time pints 24th and 72nd h) for all non-irradiated cells, which was followed by an increase in the expression, after irradiation, for all of the strains. The effect of an increased expression after irradiation was no longer observed in the chromatin mutants when the irradiation time was increased to 30 min (Figure 8B). An exception to this trend was the result on the 72nd h of cultivation of the WT, which had decreased the *ATG18* mRNA levels after irradiation. We assume that the role of only one protein is not enough to trigger the process of autophagy in our system, but rather accept these data as initial steps to prove it. Moreover, there are data that prove the importance of certain markers for the process and they could be a good background in our future research on the subject [55].

*CDC28* gene expression was studied in the process of CLS of the yeast cells and results are shown in Figure 9. Cdc28p is a master cell cycle regulator, that forms different complexes with various types of cyclins throughout the separate cell cycle phases, and is needed for the cell cycle to progress [56]. The highest levels of *CDC28* mRNA were seen in all the four strains in their stationary phase cells—72nd h and 9th day of cultivation, for both controls and irradiated for 3 min samples (Figure 9). Particularly for the 9th day of cultivation, all the three chromatin mutants showed a higher *CDC28* mRNA level after irradiation, and especially the arp4 and double mutants, having a fold change of 275 and 265, respectively, compared to a fold change of 203 for the WT. Increasing the UVA/B dose again (Figure 9B) resulted in a very abnormal gene expression for the chromatin mutants at their 9th day of cultivation, however in an inverse trend, having an average fold change difference of 428 between the *CDC28* mRNA levels of the chromatin mutants and the WT.

The resilience of the studied yeast cells to the two different doses of UVA/B stress was expected to be different. As seen in Figure 3 (CFU) and Figure 4 and Figure 5 (FACS cell cycle data analyses), the WT and the chromatin mutants have different growing abilities. It is also supposed that the response of the chromatin mutants to UVA/B stress will be accompanied by a characteristic expression of stress responsive genes that differs from that of the WT. The three genes that we have analyzed in this context, *RAD9, CDC28* and *ATG18,* were deliberately chosen as their activity reflects the way cells deal with DNA damage caused by UV light, cell cycle regulation, and autophagy promotion. Differences in the expression of these genes in the WT and mutants detected in the current work indicate that the regulation of the yeast cell response to UVA/B in the timeline of the CLS is distinct for each chromatin mutant, thus proving the role of the underlying chromatin structure in the process of CLS and UVA/B stress resilience.

## 4. Conclusions

Based on the results we achieved, it is clear that the chromatin organization affects how cells respond to stress stimuli. The linker histone has a major role in the preservation of the genome stability; it is important for the chromatin organization, as well as for the way cells respond to aging, concerning the process of remodeling their chromatin structure [27,33,37,38]. What we have seen during our experiments was that the linker histone mutant experienced increased viability after a low dose of UVA/B irradiation. It is well known that the linker histone is an inhibitor of homologous recombination. Both homologous recombination and non-homologous end-joining are special processes, used by the cells, for the repair of DNA double-strand breaks (DSBs). Exposure to UVA/B light is a common reason for the occurrence of such breaks. The lack of the gene for the linker histone, therefore, allows for a hectic repair of the caused DSBs by homologous recombination, which is beneficial regarding increased cell viability of the yeast linker histone mutant [47]. It was interesting to note that despite this ability of the linker histoneless mutant, when the *arp4* mutation was added, the double mutant was no longer able to activate this hectic DSB repair mechanism. In general, it was not able to survive well after the exerted stress and even lost its viability prematurely. This only goes on to support the fact that global chromatin reorganization is executed during the process of aging [41,57] and therefore correct chromatin organization is crucial for the cells to age properly [27].

As noted, the expression of different stress-responsive genes was examined. The linker histone is an important part of DNA damage response and maintenance of genome stability [25]. The *RAD9* gene is a regulator of the G2/M cell cycle checkpoint. A previous study [54] has shown that the *RAD9* gene is essential for the repair of the DNA damage, caused by UV light, and is characterized by the activation of the nucleotide excision repair pathway. Another gene associated with the cells being blocked at the G2/M checkpoint is *CDC28*. Its expression is typically increased, followed by UV irradiation [58]. As already mentioned, an occurred DNA damage results in the activation of DNA damage response (DDR) and thereafter cell cycle arrest. Other authors [59] have discovered an upregulation of autophagy following DNA damage, as it is being activated by the DDR. This has also resulted in the cells being prevented from entering mitosis. What they hypothesized was that to restrain the cell cycle progression, the levels of DNA repair enzymes and different proteins, promoting mitosis, were being regulated by autophagy. *ATG18* was found to be an essential gene required for the activation of autophagy, and more precisely to a DDR specific type of autophagy called genotoxin-induced targeted autophagy (GTA). The expression of all these genes was studied in our strains. Interestingly, regarding all the genes involved in the response after irradiation with higher doses of UVA/B light, our chromatin mutants were unable to properly activate these genes, resulting in much lower mRNA levels compared to the WT. This in turn indicates the importance of proper chromatin organization for the correct activation of key mechanisms that are needed for the cells to deal with UVA/B stress upon chronological aging.

## Figures and Tables

**Figure 1 cells-10-01755-f001:**
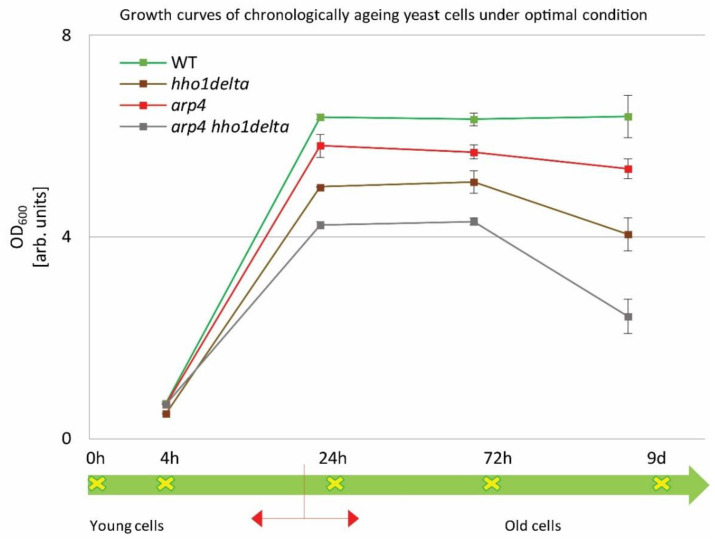
Growth curves of WT, *hho1delta*, *arp4,* and *arp4 hho1delta* yeast cells cultured for a period of nine days under optimal conditions.

**Figure 2 cells-10-01755-f002:**
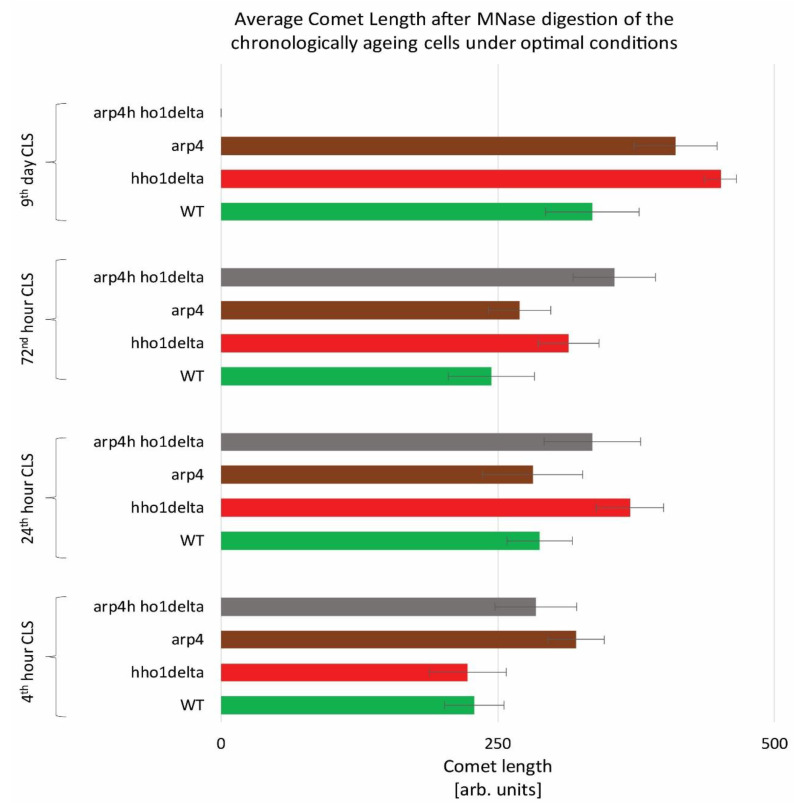
Higher-order chromatin loop dynamics of WT, *hho1delta*, *arp4,* and *arp4 hho1delta* yeast cells cultured for a period of nine days under optimal conditions assessed by ChYCA. Three repetitions of the experiment were done and results are presented as MEAN of the comet length values in (arb. units) ± STDV.

**Figure 3 cells-10-01755-f003:**
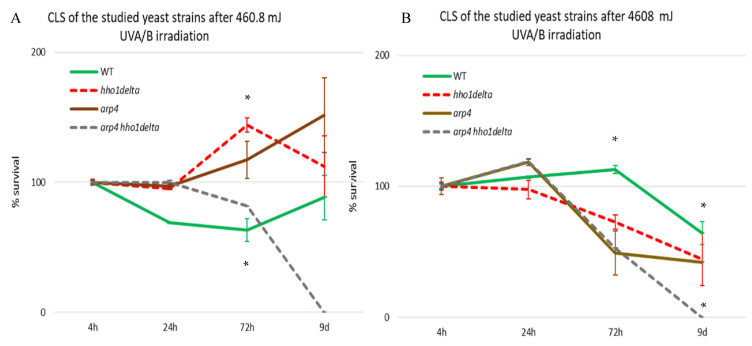
CLS of WT, *hho1delta*, *arp4,* and *arp4 hho1delta* yeast cells cultured for a period of 9 days after 3 and 30 min UVA/B irradiation. (**A**) CLS of the four studied strains after 3 min UVA/B irradiation (460.8 mJ). (**B**) CLS after 30 min UVA/B irradiation (4608 mJ). Cell survival in % is calculated for each of the four strains with the 4th hour of the WT accepted as 100% and the other cell survival time points were calculated as a percentage of it. Three repetitions of the experiment are done and results are presented as MEAN ± STDV. * designates *p* < 0.001.

**Figure 4 cells-10-01755-f004:**
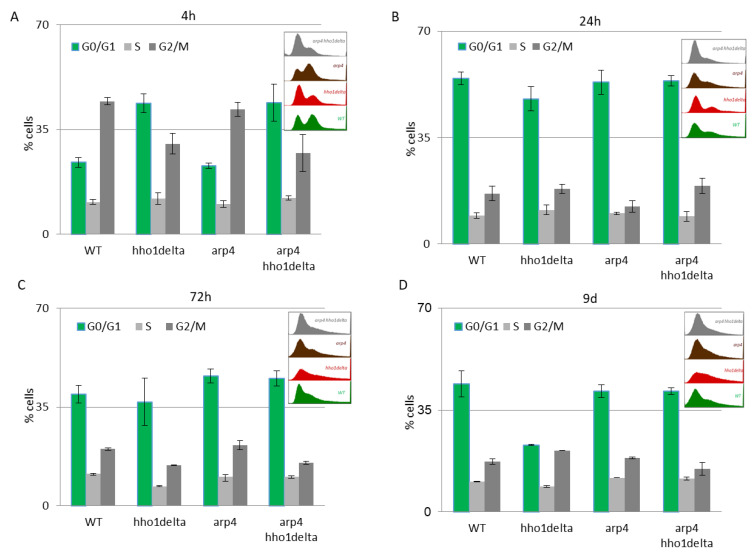
FACS data for studying the cell cycle progression of WT, *hho1delta*, *arp4,* and *arp4 hho1delta* yeast cells cultured for a period of nine days under optimal CLS conditions. Three repetitions of the experiment are done and results are presented as MEAN ± STDV. (**A**) FACS data of logarithmically growing cells with embedded representative histograms. (**B**) FACS data of 24-h yeast cell cultures with embedded representative FACS cell cycle data histograms. (**C**) FACS data of 72-h yeast cell cultures with representative FACS cell cycle data histograms. (**D**) FACS data of 9-day yeast cell cultures with representative FACS cell cycle data histograms.

**Figure 5 cells-10-01755-f005:**
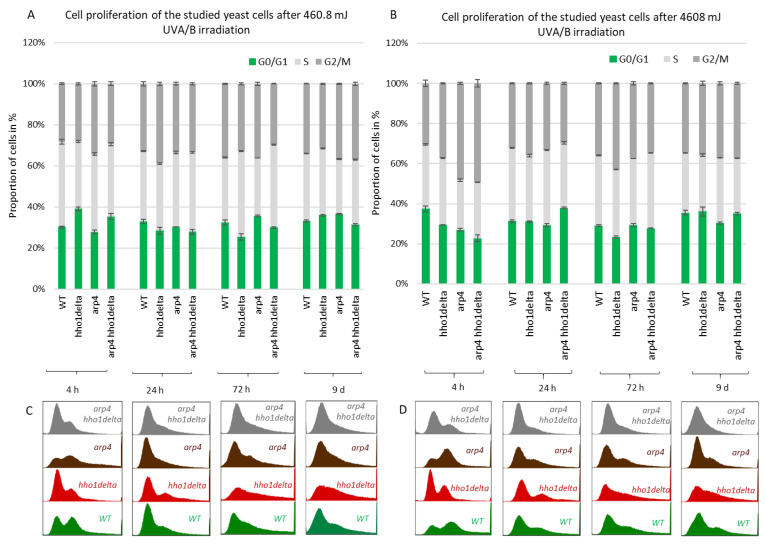
Cell cycle progression of WT, *hho1delta*, *arp4* and *arp4 hho1delta* yeast strains cultures after UVA/B irradiation. (**A**) 3 min (460.8 mJ), (**B**) 30 min (4608 mJ) UVA/B irradiation with representative FACS cell cycle histograms (**C**) for 460.8 mJ and (**D**) for 4608 mJ. Three repetitions of the experiment are done and results are presented as a percentage of the cells of each strain at the given time point without irradiation with the respective STDVs. Each color represents the portion of cells in the cell cycle phase calculated from the non-irradiated sample.

**Figure 6 cells-10-01755-f006:**
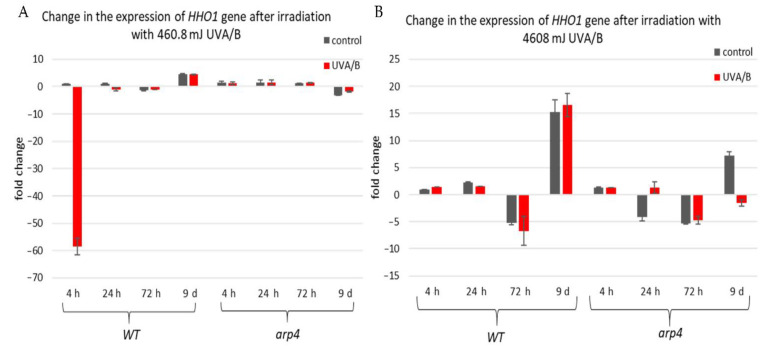
Change in the expression of *HHO1* gene after irradiation with low- and high dose UVA/B. (**A**) 3 min (460.8 mJ). (**B**) 30 min (4608 mJ). Three repetitions of the experiment were performed and results are MEAN fold change ± STDV.

**Figure 7 cells-10-01755-f007:**
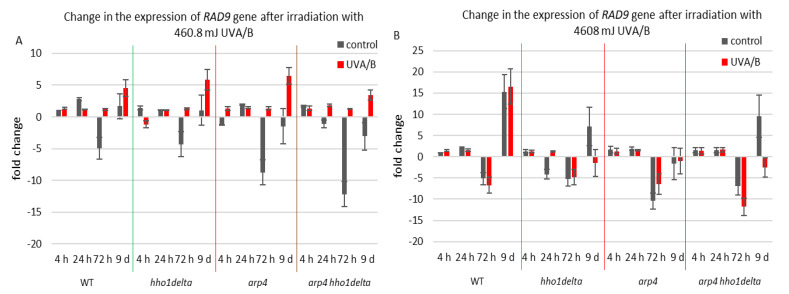
Change in the expression of *RAD9* gene after irradiation with 3 min. (**A**) (460.8 mJ). (**B**) 30 min (4608 mJ). Three repetitions of the experiment were performed and results are MEAN fold change ± STDV.

**Figure 8 cells-10-01755-f008:**
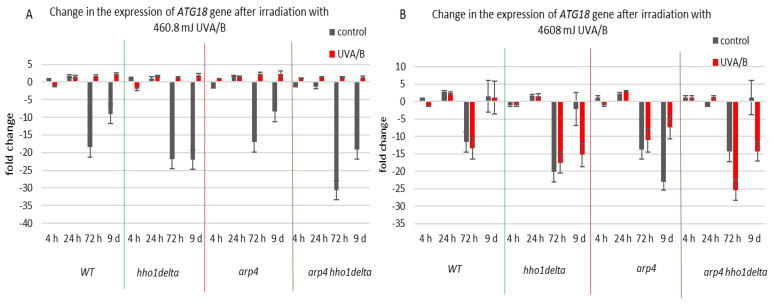
Change in the expression of *ATG18* gene after UVA/B irradiation. (**A**) 3 min (460.8 mJ). (**B**) 30 min (4608 mJ). Three repetitions of the experiment are done and results are MEAN fold change ± STDV.

**Figure 9 cells-10-01755-f009:**
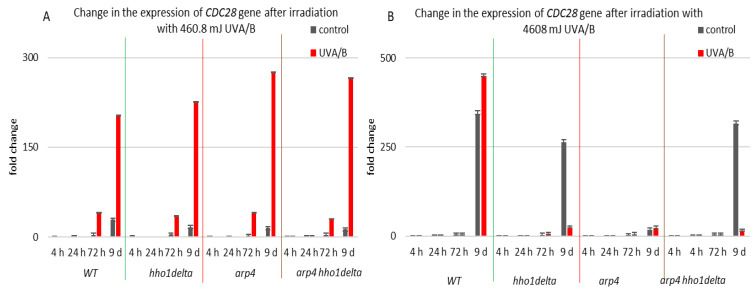
Change in the expression of *CDC28* gene after UVA/B irradiation. (**A**) 3 min (460.8 mJ). (**B**) 30 min (4608 mJ). Three repetitions of the experiment are done and results are MEAN fold change ± STDV.

## Data Availability

All data are freely available.

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
