# Peer review of "Changes in Chromatin Organization Eradicate Cellular Stress Resilience to UVA/B Light and Induce Premature Aging"

_cells, 2021, doi:10.3390/cells10071755_

Round 1

Reviewer 1 Report

The aim of this work was to investigate the role of the yeast linker histone Hho1 and the Arp4 protein in mediating the response to UV DNA damage events during chronological aging in yeast. The authors were purporting to use yeast as a model organism to gain insight into how similar processes in human cells may function during cellular aging. By using single and double yeast mutants, the authors concluded that Hho1 and Arp4 were responsible for organizing chromatin during chronological aging and that failures in this process (due to the mutations in HHO1 and ARP4) made cells more sensitive to UV damage which ultimately promoted premature aging.

Whilst the authors did show some interesting results, the study seemed to lack focus and the data was some-what over-interpreted. Furthermore, the literature cited needs to be up-dated, particularly concerning what is now known about two cell populations forming during yeast chronological aging. Unfortunately, I felt that the author’s claims were not supported by their data which I thought was preliminary.

Specific comments:

What is the nature of the arp4 mutant? I was unclear if this was a conditional mutant or not. The genotype listed in the materials and methods suggested it was. However, there was no description of cells being shifted to a non-permissive temp to gain the null mutant phenotype. I felt that this needs to be clarified.

Figure 1 monitors OD at the given time points. This graph was not measuring growth rates as stated. Indeed, growth (in terms of cell proliferation) would be severely limited by 72h and should be absent at day 9. Any changes in OD could be due to numerous differences in cell morphology. I would suggest that the authors would have been better served by counting cells under the microscope to attain a cell density at each time point together with plating to gain CFUs/ml. Together, this would yield the total cells at each time point and might indicate how many of the cells are dead or alive. This information was fundamental to the study, but was absent (although partially re-visited in Fig 3 after the UV irradiation).

The changes in chromatin structure claimed in their mutants were based on a single type of analysis called the COMET assay (Figure 2). Although, I have no reason to doubt the validity of the assay and the results, I would liked to have seen evidence of the changes in chromatin structure confirmed by some other, more established, techniques. For example the authors could have examined the sensitivity of the chromatin in their mutants via MNase digestion and subsequent gel electrophoresis. In addition, it might have been pertinent to examine histone protein levels in their mutants over time to determine if there was a loss in histone levels in their cells during chronological aging. This would have supplemented their COMET assay data nicely to give readers who are unfamiliar with this assay more confidence in the chromatin structural changes claimed.

The data shown in Fig. 4 was the most interesting data and should have been analysed further. I liked this data. In my opinion it looks like the hho1 mutant is not showing the expected hallmarks of cells in deep stationary phase. This could be either the G1 arrest is slipping post-diauxie, or the quiescent cells are not forming or are not surviving. Also, what is ‘proliferation potential’ (please define) and how can the authors be measuring proliferation rates in Day 9 stationary phase when there should be no cell proliferation?

The gene transcription analysis was very selective. I was unclear of the significance of this data to the overall story.

Central to the authors claim was that chromatin organisation affects the cells response to UV. However, the authors did not show this; a causal link between the chromatin changes they showed with a response to DNA damage was not established. The authors simply observed some aspects effecting chromatin, cell survival responses to UV irradiation, and showed some changes in transcription at selected genes. The authors did not show any direct role for the chromatin structure changes detected in their mutants affecting DNA damage repair. Indeed, neither DNA damage nor its repair were directly measured at all in this paper.

Author Response

We are very much grateful to the reviewer for all comments. They allowed us to perceive some data that have been omitted to be discussed. We have addressed all comments.

Please, find our detailed reply to all comments and remarks.

Comment

“Whilst the authors did show some interesting results, the study seemed to lack focus and the data was some-what over-interpreted. Furthermore, the literature cited needs to be up-dated, particularly concerning what is now known about two cell populations forming during yeast chronological aging. Unfortunately, I felt that the author’s claims were not supported by their data which I thought was preliminary.”

Reply

We have tried to set the focus of our work by pointing at the fact that chromatin studies in regard to CLS are of high importance and that the use of yeast strains with abrogated chromatin structure are indeed a golden standard for monitoring the role of chromatin in cellular ageing.

The last is done by adding some explanatory text together with relevant references that underlie the importance of chromatin stability and compaction for the healthy lifespan.

The literature regarding the CLS of yeast has been updated. Please see references: 12-18, as well as lines 55-70.

Though as the reviewer says our data are preliminary, we find them crucial for understanding the way yeast cells with abrogated chromatin structure age chronologically especially under UVA/B stress conditions. This allows us to insist on its publication in Cells, MDPI. Moreover, they are the logic continuation of our previous work aiming to prove the role of chromatin in ageing as cited in the current manuscript.

Specific comments:

Comment

“What is the nature of the arp4 mutant? I was unclear if this was a conditional mutant or not. The genotype listed in the materials and methods suggested it was. However, there was no description of cells being shifted to a non-permissive temp to gain the null mutant phenotype. I felt that this needs to be clarified.”

Reply

The nature of arp4 mutant has been discussed with all references supporting this. Please see lines: 124-128 in the revised manuscript. We have omitted the description of this strain as we have previous work with them, but the reviewer has right. Therefore, we have included the required data here.

Comment

Figure 1 monitors OD at the given time points. This graph was not measuring growth rates as stated. Indeed, growth (in terms of cell proliferation) would be severely limited by 72h and should be absent at day 9. Any changes in OD could be due to numerous differences in cell morphology. I would suggest that the authors would have been better served by counting cells under the microscope to attain a cell density at each time point together with plating to gain CFUs/ml. Together, this would yield the total cells at each time point and might indicate how many of the cells are dead or alive. This information was fundamental to the study, but was absent (although partially re-visited in Fig 3 after the UV irradiation).”

Reply

The optical density of the yeast culture is a good indication of how cells grow. These data are indicative for the way cells grow and a prerequisite first step in all yeast CLS studies as already reported in our published works (Miloshev, G., et al., Linker histones and chromatin remodelling complexes maintain genome stability and control cellular ageing. Mech Ageing Dev, 2019. 177: p. 55-65.).

We have corrected the word “growth rate” elsewhere in the text with only “growth”. Cells have been counted by microscope but before the experiments for cell survival under CLS (Fig.3). The reviewer has right that these data are crucial and we refer to them by citing our previous work in MAD (Miloshev, G., et al., Linker histones and chromatin remodelling complexes maintain genome stability and control cellular ageing. Mech Ageing Dev, 2019. 177: p. 55-65.) and by editing the whole text describing Figure 1 (i.e. the OD600). Please see lines 203-255.

Comment

The changes in chromatin structure claimed in their mutants were based on a single type of analysis called the COMET assay (Figure 2). Although, I have no reason to doubt the validity of the assay and the results, I would like to have seen evidence of the changes in chromatin structure confirmed by some other, more established, techniques. For example, the authors could have examined the sensitivity of the chromatin in their mutants via MNase digestion and subsequent gel electrophoresis. In addition, it might have been pertinent to examine histone protein levels in their mutants over time to determine if there was a loss in histone levels in their cells during chronological aging. This would have supplemented their COMET assay data nicely to give readers who are unfamiliar with this assay more confidence in the chromatin structural changes claimed.”

Reply

The reason why we have decided to use the Chromatin Yeast Comet Assay is because we aimed at studying the way our strains changed the organization of their loop chromatin compaction during ageing. This method is indeed very sensitive and allows detection of subtle changes in chromatin loop sizes on the level of single cells. We did not aim at studying the nucleosome repeat length nor the histone protein levels. These data are already published in our previous works that studied the role of the yeast linker histone and its interactions with chromatin remodeling complexes in chromatin organization (Georgieva, M., et al., The linker histone in Saccharomyces cerevisiae interacts with actin-related protein 4 and both regulate chromatin structure and cellular morphology. Int J Biochem Cell Biol, 2015. 59: p. 182-92.; Georgieva, M., et al., Hho1p, the linker histone of Saccharomyces cerevisiae, is important for the proper chromatin organization in vivo. Biochim Biophys Acta, 2012. 1819(5): p. 366-74.).

At this point we plan to perform detailed studies on the chromatin’s role in ageing in our future experiments but not now. The reasons for this are several: chromatin studies need big financial resource and more time. Therefore, they are in the plan for our future projects and work plans.

Though, we are very much grateful to the reviewer for this suggestion and we shall perform these studies in the near future.

Comment

The data shown in Fig. 4 was the most interesting data and should have been analysed further. I liked this data. In my opinion it looks like the hho1 mutant is not showing the expected hallmarks of cells in deep stationary phase. This could be either the G1 arrest is slipping post-diauxic, or the quiescent cells are not forming or are not surviving. Also, what is ‘proliferation potential’ (please define) and how can the authors be measuring proliferation rates in Day 9 stationary phase when there should be no cell proliferation?

Reply

We have revised all FACS data analyses and have taken into account all comments by the reviewer. His/her observation on the proliferation of hho1delta cells is of great importance for us as we have observed it before but never have the conviction to state it. The reviewer has granted us with this and this allowed a more detailed and yet better representation of FACS data. Please see lines: 358 onward.

Comment

“The gene transcription analysis was very selective. I was unclear of the significance of this data to the overall story.”

Reply

We have reanalysed and enriched our transcription analyses with descriptive text and relevant references.

Comment

Central to the authors claim was that chromatin organisation affects the cells response to UV. However, the authors did not show this; a causal link between the chromatin changes they showed with a response to DNA damage was not established. The authors simply observed some aspects effecting chromatin, cell survival responses to UV irradiation, and showed some changes in transcription at selected genes. The authors did not show any direct role for the chromatin structure changes detected in their mutants affecting DNA damage repair. Indeed, neither DNA damage nor its repair were directly measured at all in this paper.”

Reply

Well, these data maybe preliminary and might require further studies we find them as important for the deeper understanding of the molecular mechanisms that underlie the interaction of chromatin with CLS under UVA/B stress.

We are very much grateful for all comments and have tried to address all of them. Though as almost all research in the field of biology of ageing we do assume that further studies are needed in order to nailed the role of chromatin in this physiological process. We do believe that our future work will address properly and concisely all comments by the reviewer.

Reviewer 2 Report

The manuscript by Vasileva et al. is to study the impact of chromatin structure/genome organization in chronological aging after irradiation.  The authors used linker histone and chromatin remodeling KO strains to examine how cells respond to UV irradiation and evaluate their molecular mechanism. Their experiments demonstrated that the compaction of chromatin is important to the process of chronological aging and to the DNA damage repair and progression of cell cycle.  As such, this manuscript reported an interesting finding and is important to the field.  This manuscript is worthy published in Cells after some minor revisions.   Please see below for the comments.

  1. The arp4 KO mutant was used in the experiment but the author fail to provide background information about the importance of this gene and why this was chosen to compare with the linker histone mutant. It would better if the author can describe this gene further in the introduction.
  2. The authors demonstrated the average comet length for each strain used in the experiment under normal conditions. It is interesting to see how hho1 mutant, arp4 mutant and double mutant showed longer comet length than WT cell.  What happened about the comet length after irradiation? The authors should also perform comet length after irradiation experiment to show these mutants’ chromatin structure are further de-compacted after irradiation. 
  3. In Figure 3, the authors showed the CLS of both arp4 and hh1delta have higher survival rate after 460.8 mJ UVA/B exposure. This situation disappeared after 4600 mJ exposure.  The authors did not discuss this observation in detail.  One important correlation can be derived from the cell proliferation rate. Due to the presentation style of Figure 5, it is difficult to interpret through the current presentation style.  Perhaps, the authors can modify Figure 5 style so that it would be easy to interpret the result.  Alternatively, the authors can examine some CLS-related gene expression.  For example, it has been shown that aged yeast has a significantly lower expression of the SIR2, SOD1 and SOD2 genes compared to young yeast.  The author can perform qRT-PCR to show these gene expression level throughout the time course after irradiation.  this might be able to explain the observation.
  4. The legend of Figure 4 is not correct. The figure is not after the irradiation. it I sunder the normal conditions.
  5. The label of Figure 9A is not correct. It should be CDC28 instead of ATG18.
  6. Line 309 “liker” should be “linker”

Author Response

We express our gratitude to all comments done by the reviewer.

All of them are addressed below: 

  1. The arp4 KO mutant was used in the experiment but the author fail to provide background information about the importance of this gene and why this was chosen to compare with the linker histone mutant. It would better if the author can describe this gene further in the introduction.

We have added all these data in the revised version of the manuscript. Please see lines: 211-218.

  1. The authors demonstrated the average comet length for each strain used in the experiment under normal conditions. It is interesting to see how hho1 mutant, arp4 mutant and double mutant showed longer comet length than WT cell.  What happened about the comet length after irradiation? The authors should also perform comet length after irradiation experiment to show these mutants’ chromatin structure are further de-compacted after irradiation. 

Well, we agree that his question is very important too and is an important aim in our future experiments. Though for this work we accepted the dynamics in the chromatin loop organization as enough to make our conclusions. Moreover, we do not expect very big reorganization in chromatin at this level of compaction after 3 and 30 min of irradiation. But we do agree that this has to be done. For now, this is in our future plans.

  1. In Figure 3, the authors showed the CLS of both arp4 and hh1delta have higher survival rate after 460.8 mJ UVA/B exposure. This situation disappeared after 4600 mJ exposure.  The authors did not discuss this observation in detail.  One important correlation can be derived from the cell proliferation rate. Due to the presentation style of Figure 5, it is difficult to interpret through the current presentation style.  Perhaps, the authors can modify Figure 5 style so that it would be easy to interpret the result.  Alternatively, the authors can examine some CLS-related gene expression.  For example, it has been shown that aged yeast has a significantly lower expression of the SIR2, SOD1 and SOD2 genes compared to young yeast.  The author can perform qRT-PCR to show these gene expression level throughout the time course after irradiation.  this might be able to explain the observation.

Indeed, these suggestions by the reviewer are of high importance. We have revised the text for the FACS data and have included some representative histograms in order to allow easier observation of the results. We hope that now the readers will be allowed to make correlation and comparison analyses easier and righter.

  1. The legend of Figure 4 is not correct. The figure is not after the irradiation. it I sunder the normal conditions.

Already corrected. Thank you!

  1. The label of Figure 9A is not correct. It should be CDC28 instead of ATG18.

Already corrected.

  1. Line 309 “liker” should be “linker”

Corrected

Reviewer 3 Report

Vasileva et al., studied “Changes in chromatin organization eradicate cellular stress resilience to irradiation with UVA/B light and induce premature aging”. The authors attempted to evaluate the role of cross-linking histones on aging. They have used S. cerevisiae as a model organism to support their hypothesis. The authors found that during the age progress, altered cell chromatin dynamics was noticed, and leads to alterations in DNA repair and autophagy functional genes. Besides, they also identified the sensitivity of yeast cells against UVA/B radiation. Overall the study is interesting. However, few additional studies are needed to support the premature results.

Comments and suggestions

In general, there are two linker histones established, H1 & H5, however, the authors describe the fifth one alone. Do the authors think that H1 is not having any role in aging?

The authors should provide the knockout study results. Both gene and protein levels of the H1 and H5.

Figures all deserve explanatory legend, including the type of statistic applied.

Few figures (5-9) are lacking “standard deviation/SEM”, which should be provided.

Figure 1: @24 h data between WT and arp4 hho1delta looks significant, please check.

Figure 2: Representative comet microscopic pictures should be provided.

Figure 4: Representative flow cytometry data pictures should be provided.

Figure 4: @72 h G2/M phase of hho1delta is significantly less than WT, please check.

Although mRNA levels are reasonable, this should be supplemented with the protein expression.

Particularly, autophagy is differentially regulated and can’t be certain on one method with one gene expression. Please refer and cite this https://doi.org/10.3390/cells9051321

Minor

Figure 6-9: X-axis labels are not legible.

Figure 9a, label: It is CDC28, not ATG18.

Author Response

We are very much grateful for all comments. Please, find bellow our replies addressing all of them.

Comments and suggestions

  • In general, there are two linker histones established, H1 & H5, however, the authors describe the fifth one alone. Do the authors think that H1 is not having any role in aging?

We are very much sure that the linker histones do take part in ageing. Our interest in the current work was set on the yeast linker histone Hho1p. Our previous studies have shown that these linker histones are crucial for higher-order chromatin loop organization during yeast CLS. Therefore, we wanted to see here how these chromatin mutants will age under UVA/B stress conditions.

  • The authors should provide the knockout study results. Both gene and protein levels of the H1 and H5.

This was done in our previous works and is already published. Please follow: https://pubmed.ncbi.nlm.nih.gov/?term=Georgieva+M+%5Bauthor%5D+and+Miloshev+%5Bauthor%5D

  • Figures all deserve explanatory legend, including the type of statistic applied.

We have provided everywhere STDVs as all our results have been analysed by Excel.

  • Few figures (5-9) are lacking “standard deviation/SEM”, which should be provided.

Already provided everywhere. Thank you for this observation.

  • Figure 1: @24 h data between WT and arp4 hho1delta looks significant, please check.

Good observation, yes.

  • Figure 2: Representative comet microscopic pictures should be provided.

Well, we have been wondering whether to provide them, but then decided that the figures are too many and another panel of comet images for so many strains and time points would be too heavy. Moreover, we have published so many Chromatin come assay images in our other manuscripts. We do not want to get redundant.

  • Figure 4: Representative flow cytometry data pictures should be provided.

We have already provided them. Thank you for this suggestion. It helped a lot for the easier comparison and FACS data representation. See Figures 4 and 5.

  • Figure 4: @72 h G2/M phase of hho1delta is significantly less than WT, please check.

Yes, you have right. Thank you!

  • Although mRNA levels are reasonable, this should be supplemented with the protein expression.

Yes, true. Though needs a lot more time and resources which we cannot afford at this time. Moreover, these are planned in our future work.

  • Particularly, autophagy is differentially regulated and can’t be certain on one method with one gene expression. Please refer and cite this https://doi.org/10.3390/cells9051321

We definitely agree. We do not state that this is autophagy, but rather our preliminary proof that something was happening at this molecular level that needs further research. Please see lines: 532. The reference proposed by the reviewer is included.

Minor

  • Figure 6-9: X-axis labels are not legible.

Corrected.

  • Figure 9a, label: It is CDC28, not ATG18.

Corrected

Round 2

Reviewer 3 Report

The authors have addressed all the comments and revised the manuscript accordingly. 

Hereby I am endorsing the manuscript for publication. 

This manuscript is a resubmission of an earlier submission. The following is a list of the peer review reports and author responses from that submission.

Round 1

Reviewer 1 Report

1) This study was just descriptive results using mutants.

2) There is no mechanism for premature ageing.

3) Similar title is found in Pubmed.

Mech Ageing Dev

. 2019 Jan;177:55-65. DOI: 10.1016/j.mad.2018.07.002. Epub 2018 Jul 17.

Linker histones and chromatin remodelling complexes maintain genome stability and control cellular ageing

George Miloshev 1, Dessislava Staneva 1, Katya Uzunova 1, Bela Vasileva 1, Milena Draganova-Filipova 2, Plamen Zagorchev 3, Milena Georgieva 4

4) In the abstract, the background is too long compared with the results parts.

5) Title contains premature-ageing but results part of the abstract did not reveal premature ageing results.

6) Please separate results and discussion.

7) Please add some study about the mechanism of premature ageing.

8) Figure 1 is missing.

9) I suggested that this study's meaning also needs to be explained by human gene name and human pathological condition.

Reviewer 2 Report

.

Reviewer 3 Report

In this manuscript, Vasileva et al study the effect of the absence of linker histone in Saccharomyces cerevisiae. They created a linker histone-free yeast strain by gene knockout (HHO1 gene) and have traced the way cells age chronologically. Authors show that yeast cells with a mutation in ARP4 gene (the actin-related protein 4) and without the gene HHO1 for the histone linker led to strong alterations in the gene expression profiles of a subset of genes involved in DNA repair and autophagy.

They also show that the yeast mutants have reduced survival upon UVA/B irradiation.

This manuscript has serious problems in data presentation which makes it very difficult to evaluate properly. Specific points are given below.

  1. What is the rationale for using strains containing ARP4 mutations separately or along with HHO1 delta? The authors give no explanations/rationale to this important part of the study.
  2. No statistical analyses were done for Fig.1. Are the differences described significant?
  3. Figures 1, 4, and 5 are missing. Due to this, all the figure references are very difficult to understand or follow.